# Early language exposure affects neural mechanisms of semantic representations

Xiaosha Wang[1,2]*, Bijun Wang[1,2], Yanchao Bi[1,2,3]*

[1]State Key Laboratory of Cognitive Neuroscience and Learning & IDG/McGovern Institute for Brain Research, Beijing Normal University, Beijing, China; [2]Beijing Key Laboratory of Brain Imaging and Connectomics, Beijing Normal University, Beijing, China; [3]Chinese Institute for Brain Research, Beijing, China

**Abstract** One signature of the human brain is its ability to derive knowledge from language inputs, in addition to nonlinguistic sensory channels such as vision and touch. How does human language experience modulate the mechanism by which semantic knowledge is stored in the human brain? We investigated this question using a unique human model with varying amounts and qualities of early language exposure: early deaf adults who were born to hearing parents and had reduced early exposure and delayed acquisition of any natural human language (speech or sign), with early deaf adults who acquired sign language from birth as the control group that matches on nonlinguistic sensory experiences. Neural responses in a semantic judgment task with 90 written words that were familiar to both groups were measured using fMRI. The deaf group with reduced early language exposure, compared with the deaf control group, showed reduced semantic sensitivity, in both multivariate pattern (semantic structure encoding) and univariate (abstractness effect) analyses, in the left dorsal anterior temporal lobe (dATL). These results provide positive, causal evidence that language experience drives the neural semantic representation in the dATL, highlighting the roles of language in forming human neural semantic structures beyond nonverbal sensory experiences.

*For correspondence:
wangxiaosha@bnu.edu.cn (XW);
ybi@bnu.edu.cn (YB)

## Editor's evaluation

This study provides important evidence regarding the development of concept representations, using functional brain imaging to compare concept structure in people with different amounts of language experience. The analyses, which are overall solid, suggest that concept representations differ as a function of childhood language experience.

## Introduction

Humans are believed to be the only species in the animal kingdom where knowledge learning can be achieved symbolically, mostly through language ('roses are red'), in addition to sensory channels (visually perceiving roses in color red) (*Gelman and Roberts, 2017*; *Perszyk and Waxman, 2018*). A common view shared by the modern neurocognitive theories of semantics is that semantic knowledge, even those acquired through language, is ultimately grounded in (nonlinguistic) sensory/motor experiences, encoded in the distributed brain areas encompassing high-level sensorimotor areas, and potential hub regions that bind such sensory-derived representations (*Barsalou, 2016*; *Binder et al., 2009*; *Fernandino et al., 2022*; *Ralph et al., 2017*; *Martin, 2016*). However, language experiences may contribute to semantic development beyond sensory experiences by facilitating or modulating categorizations by the nature of labeling (i.e. words) to the sensory experiences, and/or by constructing semantic relations based on various types of word relations (*Gelman and Roberts, 2017*;

**eLife digest** Humans are the only known species where much of knowledge learning happens symbolically through language, in addition to information received directly from the senses. For example, humans can learn about the color of some rose flowers from the popular expression "roses are red" without needing to see any red roses – allowing them to accumulate knowledge beyond the constraints of their own senses.

Recent work suggests that a region of the brain known as the dorsal anterior temporal lobe represents knowledge acquired from language instead of sensory experiences. However, these studies were based on volunteers deprived of sensory experiences rather than those with reduced language exposure. Therefore, it was not clear whether this brain structure represents knowledge derived specifically from language and the importance of language in shaping non-sensory knowledge.

To address this question, Wang et al. studied the brain activity of deaf adult volunteers in a word meaning judgement task. Volunteers were either born deaf or lost their hearing as toddlers, and all primarily used Chinese Sign Language for communication. One group of volunteers had been exposed to sign language from birth, giving them similar exposure to language as hearing individuals. The other group had less exposure to language in their early years and only learned sign language later in childhood.

The task included 90 written words that were familiar to the volunteers. They included a mixture of object words – related to material objects – such as "shoulder" and "hammer" and abstract words – which are not linked to physical objects – such as "cause" and "violence". The volunteers were shown each word in turn and asked to think about the word's meaning. Brain scans revealed that the left dorsal anterior temporal lobes of the volunteers with reduced early language exposure were less sensitive to the meaning of the words compared with those of the other volunteers.

The findings demonstrate that the dorsal anterior temporal lobe specifically supports meaning derived from a person's experience of language as opposed to sensory experience, providing a new angle to understand the mechanism of knowledge representations. Increased understanding of how language supports knowledge will help to uncover the human-specific ways of representing and creating knowledge in the brain.

---

*Perszyk and Waxman, 2018*; *Unger and Fisher, 2021*). Are such cognitive contributions manifested by modulating the neural representations of the sensory-derived semantic spaces, or by also formulating neural representations that specialize in representing knowledge derived from language experience (i.e. not nonlinguistic sensory experience)?

Neural representations of fully nonsensory, language-derived knowledge have only recently been inferred based on studies with sensory-deprived individuals (*Bottini et al., 2020*; *Striem-Amit et al., 2018*; *Wang et al., 2020*, see *Bi, 2021*, for a review). Congenitally blind individuals, who cannot acquire visual-specific knowledge (e.g. color) through sensory experiences, can nevertheless acquire semantic structures about such knowledge behaviorally similar to those in the sighted, presumably derived from language. Such nonsensory semantic structures in the blind (and also sighted) are represented in the dorsal anterior temporal lobe (dATL), with sighted individuals additionally representing the highly similar knowledge structure (presumably derived from sensory experience) in the visual cortex (*Wang et al., 2020*). This dATL cluster is distinct from the central 'amodal' semantic hub proposed to bind together multiple sensory attributes, which is located in the more ventral-medial territory of the ATL (*Ralph et al., 2017*; *Patterson et al., 2007*; see discussions in *Striem-Amit et al., 2018*). The left dATL is also more strongly activated by abstract than concrete words in typically developed individuals (*Binder et al., 2009*; *Bucur and Papagno, 2021*; *Wang et al., 2010*; *Wang et al., 2019*). It was thus proposed that this area represents knowledge derived from language, in addition to those from sensory experiences in various perceptual (association) regions (*Bi, 2021*). These lines of evidence, while highly suggestive, are based on manipulating sensory experience by examining individuals deprived of a sensory modality or by contrasting concepts with rich sensory experiences versus those without, rather than the positive manipulation of language experience. Thus, the positive evidence for the necessity of language experiences for the neural semantic representation here is still lacking.

Here, we address this issue in a special human model that varies in early language exposure: individuals who were born profoundly deaf in hearing families and had limited natural language exposure (speech or sign) during early childhood (*Goldin-Meadow and Feldman, 1977*; *Mayberry et al., 2002*). These individuals often acquired their first language (sign language) around school age, which is much later than the age of first language (L1) acquisition in typically developed children or in native deaf signers who were born in deaf families and acquired sign language from birth. Previous research has shown that such early language exposure limitation led to long-lasting effects on various aspects of language processing. Behaviorally, these delayed deaf signers, even during adulthood with many years of sign language usage, have lower proficiency in phonological, morphological, and syntactic processing of sign language (*Bogliotti et al., 2020*; *Caselli et al., 2021*; *Cheng and Mayberry, 2021*; *Lieberman et al., 2015*; *Mayberry et al., 2002*; *Mayberry and Fischer, 1989*; *Newport, 1990*; *Tomaszewski et al., 2022*). Neurally, brain functional imaging studies have reported decreased activation magnitude in the left inferior frontal and posterior temporal regions in tasks of sign sentence judgments (*Mayberry et al., 2011*; *Richardson et al., 2020*; *Twomey et al., 2020*). Anatomical alterations in regions typically recruited in language tasks – reduced cortical volume in the left inferior frontal region, reduced cortical thickness in the left posterior middle temporal region, and reduced fractional anisotropy values in the left arcuate fasciculus – were also reported in signers with delayed L1 acquisition (*Cheng et al., 2023*; *Cheng et al., 2019*).

Despite these documented effects of delayed L1 acquisition on phonology, morphology, and syntax (*Curtiss, 1977*; *Lenneberg, 1967*; *Lillo-Martin and Henner, 2021*; *Mayberry and Kluender, 2018*), studies have reported little effects on semantic behaviors, including semantic interference effects in the picture-sign paradigm (*Baus et al., 2008*), scalar implicature (*Davidson and Mayberry, 2015*), or accuracy scores of several written word semantic tasks (e.g. synonym judgment) (*Choubsaz and Gheitury, 2017*). However, as shown by the color knowledge in the congenitally blind studies (e.g. *Wang et al., 2020*), similar semantic behaviors may arise from (partly) different neural representations. Semantic processing is supported by a multifaceted cognitive system and a complex neural network entailing distributed brain regions (*Bi, 2021*; *Binder and Desai, 2011*; *Ralph et al., 2017*; *Martin, 2016*), and thus focal neural changes may not necessarily lead to semantic behavioral changes. Neurally, neurophysiological signatures assumed to reflect semantic processes showed incongruent effects across studies: N400 effects in the semantic violation of written sentences were not affected (*Skotara et al., 2012*), whereas M400 in the picture-sign matching task showed atypical activation patterns (reduced recruitment of the left fronto-temporal regions and involvement of the right parietal and occipital regions) (*Ferjan Ramirez et al., 2016*; *Ferjan Ramirez et al., 2014*; *Mayberry et al., 2018*). It remains to be tested whether and where delayed L1 acquisition affects how semantics are neurally represented, using imaging techniques with higher spatial resolutions.

In this study, by a rare opportunity of manipulation of early language exposure offered by nature, we aim to test the neural system representing the language-derived semantic representations, beyond the sensory-derived semantic representations (*Bi, 2021*; *Wang et al., 2020*). With fMRI experiments on semantic processing of familiar words, we compared the neural semantic structures between congenitally deaf signers with delayed sign language acquisition (delayed deaf signers) and congenitally deaf signers with native sign language acquisition (native deaf signers), that is, two groups that are matched on their nonlinguistic sensory experiences but varied in early language experience. Significant group differences would provide positive evidence for the role of language experience in the formation of the identified neural semantic structures.

## Results
### Participants' background information and task fMRI design
We recruited two adult groups of congenitally or early deaf participants, including 16 native deaf signers and 23 delayed deaf signers (*Table 1*; *Table 1—source data 1*). Native signers were born in deaf families; delayed signers were born in hearing families and became exposed to Chinese sign language (CSL) between the ages of 4 and 10 (mean ± standard deviation [SD]: 6.91±1.62 years of age). Note that in China the nation-wide hearing screening for newborns or during early infancy started in 2009 and our participants were born before 2000. The family signing environment was confirmed by the subjective ratings of parental CSL proficiency: Native signers rated their parents

**Table 1.** Background demographic and language information of native and delayed signers.

| | Native (n=16) | Delayed (n=23) | Welch's t | p-Value | Hedges' g |
|---|---|---|---|---|---|
| AoA of CSL (years of age) | 0±0 | 6.91±1.62 | | | |
| Parental CSL proficiency (1–7 scale)* § | 6.81±0.54 | 2.74±1.29 | 13.54 | <0.001 | 4.04 |
| *Demographic information* | | | | | |
| Gender | 11 M, 5 F | 12 M, 11 F | | | |
| Age (years) | 28.50±7.13 | 27.09±5.87 | 0.65 | 0.52 | 0.21 |
| Education (years) | 14.13±2.31 | 15.09±1.41 | –1.49 | 0.15 | –0.49 |
| McArthur Scale of Subjective Social Status, childhood (1–10 scale)† | 3.88±2.58 | 4.35±1.97 | –0.62 | 0.54 | –0.20 |
| Father's education (years) § | 7.19±3.04 | 10.39±2.84 | –3.33 | 0.002 | –1.07 |
| Mother's education (years) | 6.56±3.14 | 8.52±2.73 | –2.02 | 0.053 | –0.65 |
| *Language information* | | | | | |
| CSL comprehension (1–7 scale)* | 6.06±0.93 | 6.17±0.89 | –0.38 | 0.71 | –0.12 |
| CSL production (1–7 scale)* | 5.75±1.00 | 5.87±1.10 | –0.35 | 0.73 | –0.11 |
| Lipreading of acquaintances (1–7 scale)* | 2.81±1.11 | 3.17±1.19 | –0.97 | 0.34 | –0.31 |
| Lipreading of strangers (1–7 scale)* | 1.88±0.81 | 1.87±0.97 | 0.02 | 0.99 | 0.01 |
| Adult Reading History Questionnaire (ARHQ, 1–5 scale)‡ | 2.59±0.68 | 2.84±0.52 | –1.25 | 0.22 | –0.41 |
| ARHQ, Item 26, self-rated reading speed (1–5 scale)‡ | 2.38±1.36 | 3.13±1.01 | –1.89 | 0.07 | –0.62 |
| ARHQ, Item 29, self-rated writing skills (1–5 scale)‡ § | 2.25±1.00 | 3.17±0.94 | –2.91 | 0.007 | –0.93 |
| ARHQ, Item 40, self-rated reading comprehension (1–5 scale)‡ | 2.44±0.81 | 2.70±0.97 | –0.90 | 0.38 | –0.28 |

*Higher scores indicate higher proficiency.

†Higher scores indicate higher social status; from **Adler et al., 2000**.

‡Higher scores indicate a higher risk of reading disability; from **Lefly and Pennington, 2000**.

§p < 0.01, significant differences between the two deaf groups.

The online version of this article includes the following source data for table 1:

**Source data 1.** Background demographic and language information of native and delayed signers.

to have much more proficient CSL skills than delayed signers did on a 7-point scale (Welch's $t_{31.7}$ = 13.54, p=1.10 × 10⁻¹⁴; each participant provided CSL ratings for both parents and the maximum score was used for group comparison). All deaf participants received formal education in special education programs since elementary school. The two deaf groups were matched on demographic variables (gender, age, years of education, ps > 0.15) and subjective social status (*Adler et al., 2000*) during childhood (p = 0.54). While the two groups significantly differed in the education levels of their parents (*Table 1*; ps < 0.053), the direction was in favor of delayed signers as their hearing parents received more formal education than the deaf parents of native signers. In terms of language skills, the two deaf groups were matched on self-reported proficiency of CSL comprehension, production, and lipreading skills (ps > 0.34), and on the reading disability risk measured by the Adult Reading History Questionnaire (ARHQ, *Lefly and Pennington, 2000*) (p = 0.22). Written word processing was further evaluated in two reaction-time tasks (i.e. visual lexical decision and word-triplet semantic judgment) and no significant group differences were found (*Figure 1—figure supplement 1*). In summary, the native and delayed deaf groups were carefully matched on a wide range of demographic and later language performances (sign, lipreading, and written). Thus, different early language exposure is a strong candidate to account for the neural group differences reported below.

The stimuli of task fMRI were 90 written words that were highly familiar to both groups, including 40 concrete/object words varying in sensory and motor properties (animals, face/body parts, artifacts) and 50 abstract/nonobject words varying in social and emotional contents. These words were grouped into 10 categories based on the group-averaged semantic multi-arrangement performances in an independent group of 32 hearing participants (see Materials and methods, *Figure 1a*, and *Figure 1—source data 1*). Both deaf groups judged these words to be highly familiar (7-point familiarity ratings from 8 native signers: 6.72±0.17; 13 delayed signers: 6.86±0.15; 7 being most familiar) and yielded similar word-relational structures in a 1 hr semantic distance judgment task, in which each participant (16 native and 21 delayed signers) produced a 90 × 90 semantic representational distance matrix (RDM). The group-averaged semantic RDMs of the two deaf groups were strongly correlated (Spearman's rho = 0.71, n = 4005; *Figure 1a*). At the individual level, correlations with the benchmark (the group-averaged RDM of hearing participants) did not significantly differ between the two deaf groups (Welch's $t_{31.3}$ = –0.37, p = 0.71; *Figure 1—source data 2*).

In the MRI scanner, participants were asked to think about the meaning of each of the 90 words (condition-rich fMRI design; *Kriegeskorte et al., 2008*) and to decide whether a word in red (catch trials) was semantically related to its previous word (i.e. oddball one-back semantic judgment, *Figure 1b*). We first examined the two-deaf-group differences in brain regions preferring abstract/nonobject words to concrete/object words, which have been proposed to relate to verbal semantic processing (*Binder et al., 2009*; *Wang et al., 2010*). These regions were functionally localized by contrasting abstract/nonobject words (e.g. *reason*) to concrete/object words (e.g. *panda*) in an independent group of 33 hearing participants (voxel-level p<0.001, cluster-level FWE-corrected p<0.05). This contrast in the hearing group resulted in regions in the frontal and temporal cortices that were well aligned with the literature (*Figure 1c*; *Figure 1—source data 3*). In particular, the left dATL, the left inferior frontal gyrus (IFG), and the left posterior middle temporal gyrus (pMTG) were most consistently reported in the meta-analyses (*Binder et al., 2009*; *Bucur and Papagno, 2021*; *Wang et al., 2010*) and were taken as our primary regions of interest (ROIs). Results for the other clusters are shown in *Figure 1—figure supplement 2* (no significant group differences). We then carried out whole-brain analyses to explore the two-deaf-group differences beyond these semantic ROIs. We focused on two neural semantic effects: (1) representational similarity analysis (RSA) of word meaning by correlating 90 words' neural RDMs with semantic RDMs (i.e. semantically related words have similar neural patterns) and (2) the univariate semantic abstractness effects.

## Semantic structure representation: dATL alteration in delayed signers

We first examined whether early-life language exposure affects neural representations of the semantic space using RSA (*Figure 2a*, *Kriegeskorte et al., 2008*). We estimated the multivoxel activation pattern for each target word and, in a given ROI, calculated the correlation distance (1-Pearson correlation) of the multivoxel activation patterns between each word pair to build a 90 × 90 neural RDM. As a sanity check, we first carried out whole-brain searchlight RSA of visual similarity of written words, by computing Spearman's rank correlation between the pixelwise dissimilarity of the 90 words

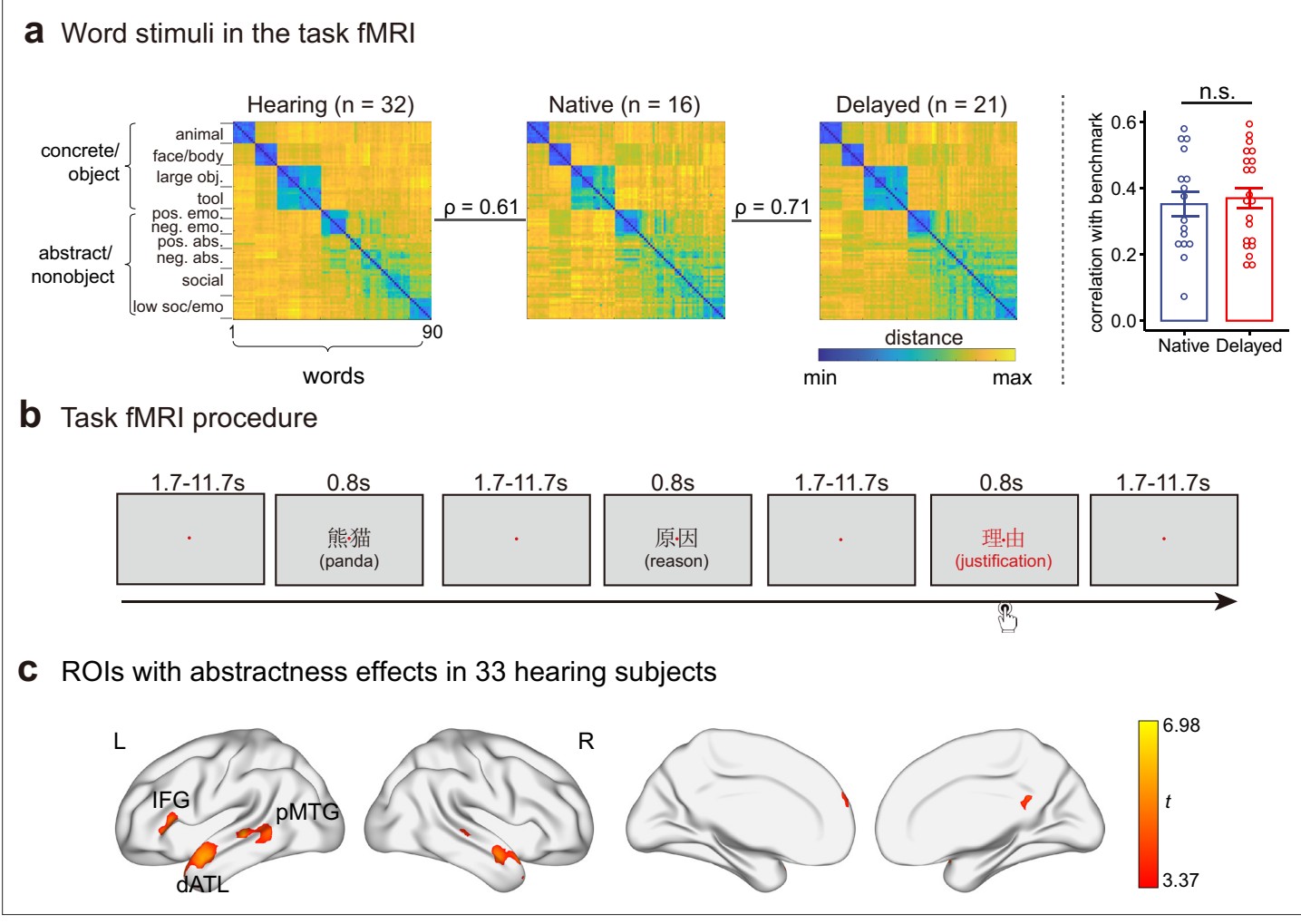

**Figure 1.** The word stimuli, task fMRI procedure, and regions of interest (ROIs) in this study. (**a**) Word stimuli in the task fMRI. Ninety words were used, including 40 concrete/object words and 50 abstract/nonobject words, which were grouped into fine-grained categories based on k-means clustering of the group-averaged semantic space of 32 hearing participants. The left panel shows the group-averaged semantic representational matrices (RDM) in hearing participants, native signers, and delayed signers. The right panel shows the Spearman's rho between each deaf participant's semantic RDM and the group-averaged semantic RDM in hearing participants. n = 16 in the native group; n = 21 in the delayed group. Error bars indicate 1 s.e.m. n.s., not significant, p > 0.05, Welch's t-test. (**b**) Task fMRI procedure. During scanning, participants were asked to think about each of 90 target word meanings (displayed in black, e.g. 'panda', 'reason') and to determine whether occasional words displayed in red (catch trials, e.g. 'justification') were semantically related to the previous word. There were 90 target word trials (each word appeared once) and 14 catch trials in each run. (**c**) Semantic ROIs were functionally identified by contrasting abstract/nonobject words with concrete/object words in 33 hearing participants at the threshold of voxel-level p < 0.001, cluster-level FWE-corrected p < 0.05. dATL, dorsal anterior temporal lobe; IFG, inferior frontal gyrus; pMTG, posterior middle temporal gyrus. L, left hemisphere; R, right hemisphere.

The online version of this article includes the following source data and figure supplement(s) for figure 1:

**Source data 1.** Ninety words in the fMRI task, grouped into 10 semantic clusters based on k-means clustering of the group-mean hearing semantic space.

**Source data 2.** Spearman's rho between each deaf participant's semantic representational distance matrix (RDM) and the group-averaged semantic RDM in hearing participants.

**Source data 3.** Details of semantic regions of interest (ROIs), including cluster location, extent, peak t values, and Montreal Neurological Institute (MNI) coordinates.

**Figure supplement 1.** Written word processing performances of the two deaf groups in two reaction-time tasks.

**Figure supplement 2.** Representational similarity analysis (RSA) and univariate results in other regions of interest (ROIs) (except for the left dATL, inferior frontal gyrus [IFG], and pMTG) in the two deaf groups.

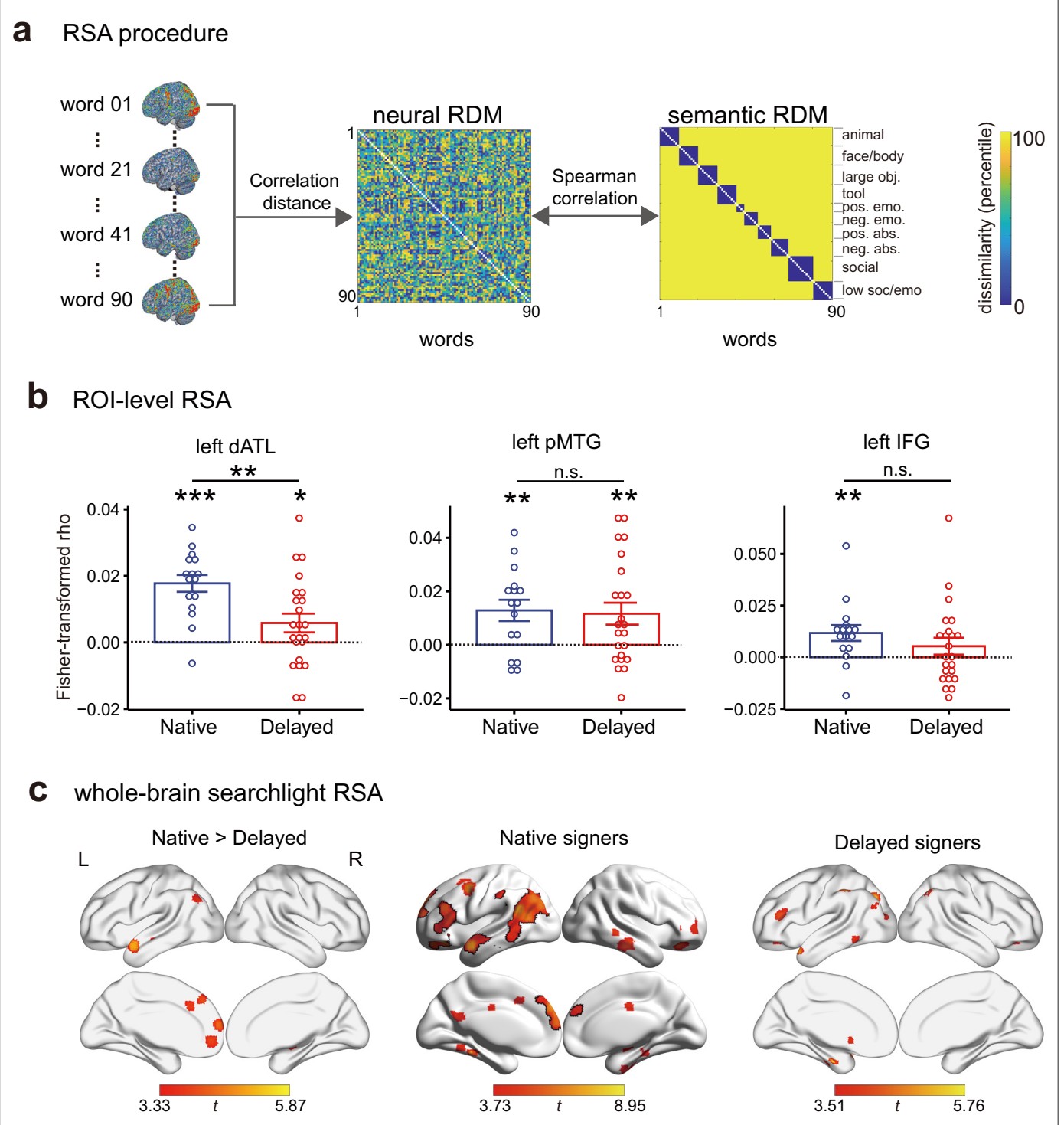

**Figure 2.** Effects of early language exposure on neural representations of semantic knowledge. (**a**) Representational similarity analysis (RSA) procedure. For each participant, a neural representational distance matrix (RDM) was computed as the correlation distance of multivoxel activity patterns (in a given regions of interest [ROI]) for each pair of words and then correlated with the hearing-group-level semantic category RDM to quantify semantic information encoded in the neural RDM. (**b**) ROI-level RSA results. Error bars indicate 1 s.e.m. n.s., not significant, $p > 0.05$; *, $p < 0.05$; **, $p < 0.01$; ***, $p < 0.001$. ROI results were assessed using one-sample t-tests (one-tailed) for each group; group differences were examined using two-tailed Welch's t-tests. dATL, dorsal anterior temporal lobe; pMTG, posterior middle temporal gyrus; IFG, inferior frontal gyrus. (**c**) Whole-brain searchlight RSA results. The statistical maps were thresholded at voxel-level $p < 0.001$, cluster size > 10 voxels. Brain results were visualized using the 'Maximum Voxel' mapping

*Figure 2 continued on next page*

*Figure 2 continued*

algorithm in BrainNet Viewer to illustrate small clusters. Clusters in black lines survived the cluster-level FWE-corrected p < 0.05. L, left hemisphere; R, right hemisphere. n = 16 in the native group; n = 23 in the delayed group.

The online version of this article includes the following source data and figure supplement(s) for figure 2:

**Source data 1.** Region of interest (ROI)-level representational similarity analysis (RSA) results in the two deaf groups.

**Source data 2.** Cluster details of the whole-brain searchlight representational similarity analysis (RSA) results.

**Figure supplement 1.** Whole-brain representational similarity analysis (RSA) results of pixelwise similarity of visual words in the two deaf groups.

and neural RDMs, and observed significant encoding of written word visual similarity in the early visual cortex in both native and delayed signers at the threshold of voxel-level p < 0.001, cluster-level FWE-corrected p < 0.05 (*Figure 2—figure supplement 1*). To quantify the semantic information encoded in the neural RDMs, in each deaf subject, we computed Spearman's partial correlation between the neural RDMs and the 10-category benchmark semantic RDM (i.e. those of the hearing group), while controlling for the stimulus low-level visual and phonological RDMs (see Materials and methods). Note that we opted for the categorical structural similarity based on the clustering analyses to boost signal and to allow for better generalization across items (i.e. along the categorical structure). This approach may lose the important graded space especially for the abstract items, and we carried out a validation analysis using the continuous semantic distances specifically focused on the abstract items (see below).

ROI-level RSA was carried out in the left dATL, pMTG, and IFG. As shown in *Figure 2*, *Figure 2—source data 1*, the three ROIs significantly encoded the semantic space in native signers ($t_{15} > 3.04$, one-tailed ps < 0.004, Cohen's d > 0.76). In delayed signers, the left dATL and pMTG also significantly encoded the semantic space ($t_{22} > 2.07$, one-tailed ps < 0.025, Cohen's d > 0.43) and the left IFG showed a similar, nonsignificant, trend ($t_{22} = 1.31$, one-tailed p = 0.10, Cohen's d = 0.27). Critically, group differences were observed in the left dATL (Welch's $t_{36.7} = 3.15$, two-tailed p=0.003, Hedges' g = 0.98), not in the left pMTG or IFG (ps > 0.26). Note that we did not observe a significant group-by-ROI interaction in the two-way ANOVA ($F_{(2,74)} = 1.59$, p = 0.21).

We then carried out whole-brain searchlight RSA (*Kriegeskorte et al., 2006*) to explore group differences in semantic encoding beyond the semantic ROIs defined above (*Figure 2c*; *Figure 2—source data 2*). At the threshold of voxel-level p < 0.001, cluster-level FWE-corrected p < 0.05, in native signers, the semantic encoding was observed in the left dATL, IFG, pMTG, along with adjacent parietal and occipital areas, dorsomedial prefrontal cortex, and frontal orbital cortex. In delayed signers, no brain regions survived the whole-brain correction; at a more lenient threshold of voxel p < 0.001, cluster size >10 voxels, semantic encoding could be observed in the left dATL, IFG, and clusters scattered in the left inferior parietal and occipital regions. Of course, such a thresholded map does not indicate true negatives in the delayed singer group. The whole-brain contrast between the two deaf groups peaked at the left dATL at the threshold of voxel p < 0.001, which was also the largest cluster (peak MNI coordinates: –60, 6,–20, peak t = 5.87, 162 voxels), converging well with the ROI results in that native signers showed stronger semantic encoding than delayed signers in the left dATL ROI. While this cluster was not large enough to survive the whole-brain correction (cluster FWE-corrected p = 0.22), the top 10 voxels survived the voxel-level correction of FWE-corrected p < 0.05. No areas were found to show significantly increased semantic information in delayed signers compared with native signers at the conventional threshold. Together, group differences were not apparent outside the ROIs analyzed in the previous section.

To further validate the reduced semantic encoding in the left dATL, we carried out the following analyses: (1) Types of semantic distance measures: While semantic categories for concrete/object words are robust and well documented, the semantic categorization within the abstract/nonobject words is much fuzzier and remains controversial (*Catricalà et al., 2014*; *Wang and Bi, 2021*). The behavioral semantic RDM in *Figure 1a* indeed shows gradations in dissimilarity for abstract/nonobject words. We thus checked the two groups' semantic RDMs using the continuous behavioral measures and further examined whether group differences in the left dATL were affected by the types of semantic distance (categorical vs. continuous) being used for abstract/nonobject words. The two deaf groups showed comparable similarities to the hearing benchmark (by correlating each deaf subject's RDM with the group-averaged RDM of hearing subjects, Welch's $t_{23.0} = -0.12$, two-tailed p = 0.90). RSA was performed

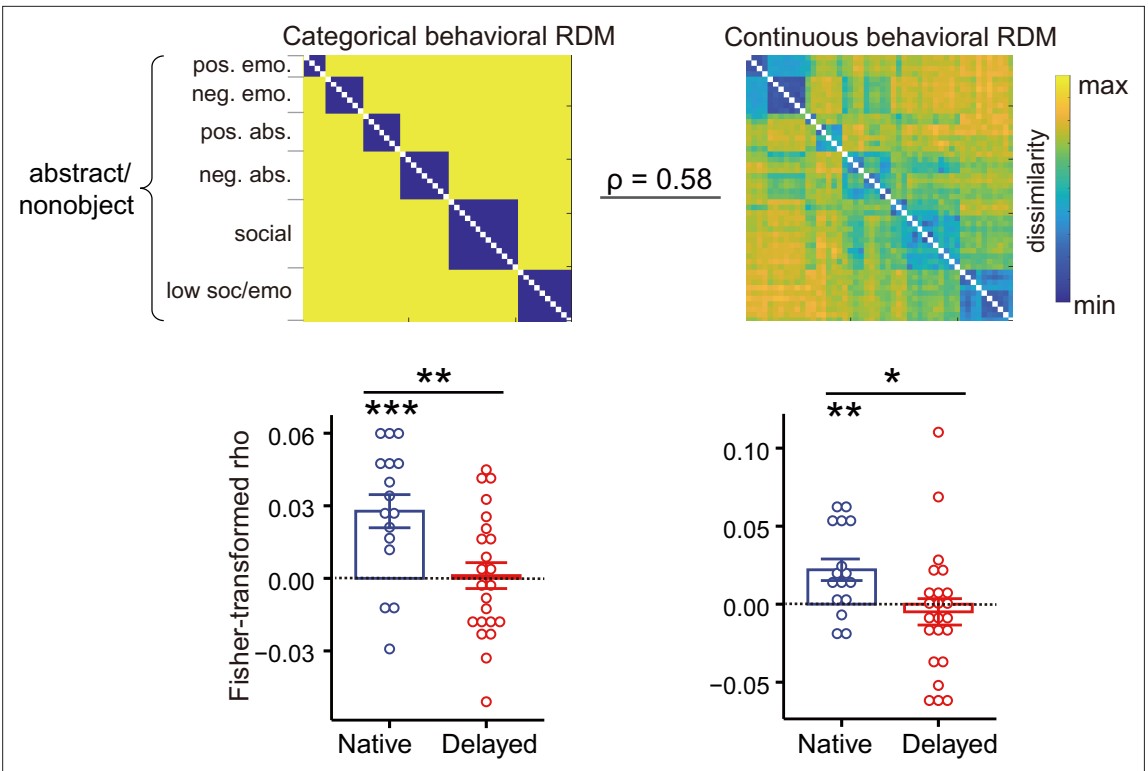

**Figure 3.** Effects of early language exposure on neural representations of abstract/nonobject words in the left dorsal anterior temporal lobe (dATL). The two representational distance matrices (RDMs) at the top illustrate the categorical and the continuous behavioral semantic RDMs of 50 abstract/nonobject words constructed in an independent group of hearing individuals. The bar plots at the bottom show the Fisher-transformed Spearman's correlations between the neural and semantic RDMs for each deaf subject. Error bars indicate 1 s.e.m. *, p < 0.05; **, p < 0.01; ***, p < 0.001. RSA results were assessed using one-sample *t*-tests (one-tailed) for each group; group differences were examined using two-tailed Welch's *t*-tests. n = 16 in the native group; n = 23 in the delayed group.

The online version of this article includes the following source data for figure 3:

**Source data 1.** Representational similarity analysis (RSA) results for the abstract/nonobject words in the two deaf groups.

by correlating each deaf subject's neural RDM in the left dATL with these two types of semantic RDMs. Significant group differences were observed (*Figure 3*; *Figure 3—source data 1*), for both the categorical RDM (Welch's $t_{31.0}$ = 3.06, two-tailed p = 0.005, Hedges' *g* = 0.98) and the continuous behavioral semantic RDM (Welch's $t_{37.0}$ = 2.47, two-tailed p = 0.018, Hedges' *g* = 0.76), with significant semantic encoding in the dATL observed in both analyses for native signers (one-tailed ps < 0.003) and neither for delayed signers (one-tailed ps > 0.42). These results indicate that the reduced dATL encoding of abstract/nonobject word meanings induced by delayed L1 acquisition was reliable across different semantic distance measures. (2) Types of ROIs: To validate whether the dATL semantic reduction in delayed signers depends on particular ATL definitions and to explore potential group differences in other language-sensitive regions beyond the ROIs we localized, we performed the RSA in a commonly used language mask (contrasting intact sentences with nonword lists) (*Fedorenko et al., 2010*). As shown in *Figure 4*, *Figure 4—source data 1*, again we observed significant group differences in the ATL (Welch's $t_{33.1}$ = 3.71, two-tailed p = 7.53 × 10⁻⁴, Hedges' *g* = 1.18), which also survived the Bonferroni correction. Other language-sensitive regions did not reach significance, with the tendency for the same direction of semantic encoding reduction (ps > 0.065, uncorrected). Two-way ANOVA showed a significant main effect of group ($F_{(1, 37)}$ = 6.80, p=0.013) and no significant ROI-by-group interaction ($F_{(5, 185)}$ = 0.823, p=0.535), indicating that delayed L1 acquisition resulted in widespread reduced semantic representations in the language regions, with the effects in the ATL consistently robust.

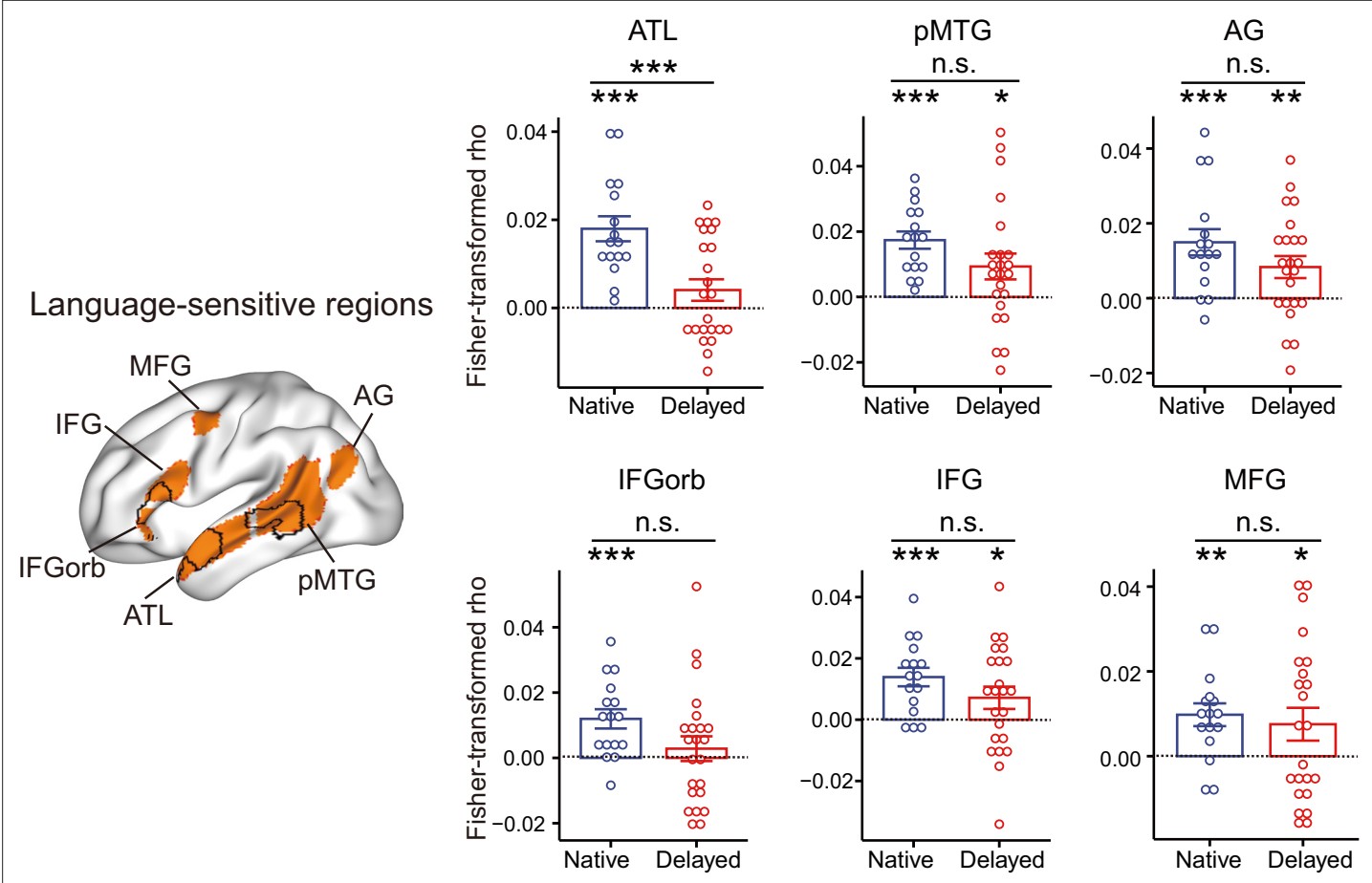

**Figure 4.** Effects of early language exposure on neural representations of semantic knowledge in language-sensitive regions. The left panel shows the language-sensitive regions of interest (ROIs) (*Fedorenko et al., 2010*) and the black lines indicate the three ROIs we functionally localized. Bar plots show the Fisher-transformed Spearman's correlations between the neural and the semantic category RDMs for each deaf subject. Error bars indicate 1 s.e.m. n.s., not significant, p > 0.05; *, p < 0.05; **, p < 0.01; ***, p < 0.001. RSA results were assessed using one-sample t-tests (one-tailed) for each group; group differences were examined using two-tailed Welch's t-tests. ATL, anterior temporal lobe; pMTG, posterior middle temporal gyrus; AG, angular gyrus; IFGorb, the orbital part of the inferior frontal gyrus; IFG, inferior frontal gyrus; MFG, middle frontal gyrus. n = 16 in the native group; n = 23 in the delayed group.

The online version of this article includes the following source data for figure 4:

**Source data 1.** Representational similarity analysis (RSA) results in language-sensitive regions of interest (ROIs) in the two deaf groups.

## Univariate semantic abstractness effects: dATL alteration in delayed signers

We then examined how early language exposure might alter the neural semantic abstractness effects by comparing regional activation strength to abstract/nonobject and concrete/object words in the two deaf groups. While the abstractness effect has often been used to reflect linguistic processes (e.g. *Wang et al., 2010*), 'abstractness' is not a single dimension and relates to both linguistic and nonlinguistic (e.g. emotion) cognitive processes (*Binder et al., 2016*; *Troche et al., 2014*; *Wang et al., 2018*). The reasoning here is that if a brain region's abstractness effect is affected by early language exposure, this would constitute further evidence that the abstractness computed by this brain region is indeed associated with language processes.

At the ROI level (*Figure 5—source data 1*), all three ROIs (the left dATL, IFG, and pMTG) exhibited significant abstractness preference in both deaf groups (*Figure 5a*; native group: paired $ts > 3.72$, df = 15, one-tailed ps <0.001, Cohen's $ds > 0.93$; delayed group: paired $ts > 2.13$, df = 22, one-tailed ps < 0.02, Cohen's $ds>0.44$), further indicating the robustness of semantic abstractness preference in these ROIs. We then carried out two-way ANOVA to examine group effects in each of these ROIs, with

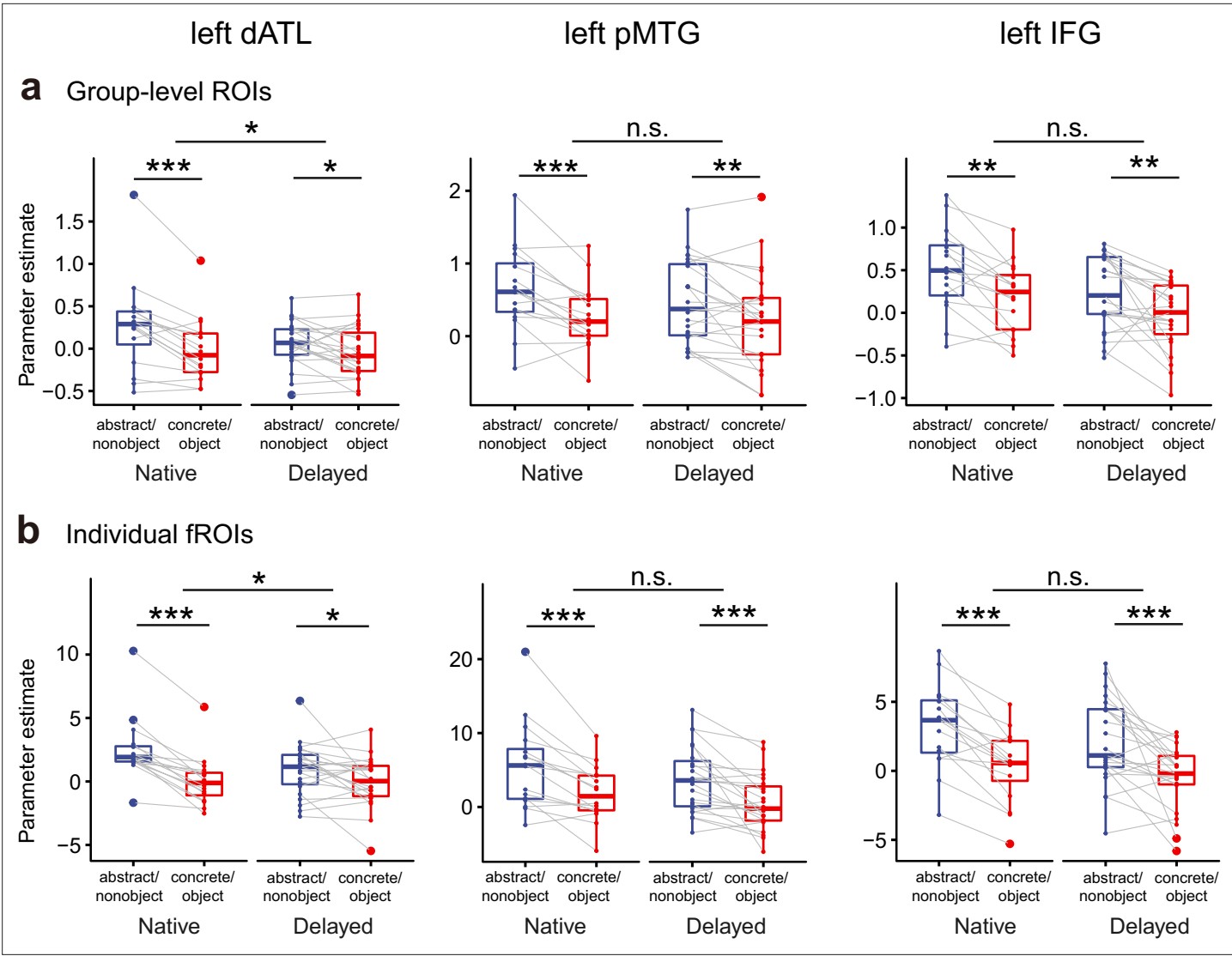

**Figure 5.** Effects of early language exposure on the univariate semantic abstractness effects in group-level-defined semantic regions of interest (ROIs) (**a**) and individual functional ROIs (fROIs) (**b**). Boxplots show beta values to abstract/nonobject words and concrete/object words in the three ROIs (the left dorsal anterior temporal lobe [dATL], posterior middle temporal region [pMTG], and inferior frontal gyrus [IFG]). n.s., not significant, $p > 0.05$; *, $p < 0.05$; **, $p < 0.01$; ***, $p < 0.001$. Beta values of the two word types were compared using one-tailed paired $t$-tests for each group; group effects were assessed using two-way analysis of variance. n=16 in the native group; n = 23 in the delayed group.

The online version of this article includes the following source data and figure supplement(s) for figure 5:

**Source data 1.** Raw beta values to abstract/nonobject words and concrete/object words in group-level-defined semantic regions of interest (ROIs) and individual functional ROIs (fROIs) in the two deaf groups.

**Figure supplement 1.** Whole-brain univariate results of semantic abstractness in native and delayed signers.

word type (concrete/object, abstract/nonobject) as a within-subject factor and group (native, delayed) as a between-subject factor. The group × word type interaction reached statistical significance only in the left dATL ($F_{(1,37)} = 4.91$, $p = 0.033$), not in the left pMTG or IFG (ps > 0.30). In the left dATL native signers exhibited greater semantic abstractness effects than delayed signers. The main effects of group were not significant in any of the three ROIs (ps > 0.057). We then compared group differences across the three ROIs using two-way ANOVA on the semantic abstractness effects (calculated by subtracting the concrete/object activation from the abstract/nonobject activation strength). NThere were no significant group-by-ROI interaction ($F_{(2,74)} = 0.50$, $p = 0.61$).

Considering inter-subject variations in activation locations, we further carried out the individualized functional ROI (fROI) analysis for validation (*Cohen et al., 2019*; *Ratan Murty et al., 2020*)

(see Materials and methods). Using each of the ROIs defined in the hearing group as anatomical constraints, we identified the top 50 selective voxels in each deaf participant using one-half of the fMRI data and computed regional activation strength to abstract and concrete words in these voxels in the held-out data (i.e. via an odd-even run cross-validation process). As shown in *Figure 5b*, the group × word type interaction was again observed in the left dATL ($F_{(1,37)}$ = 6.20, p = 0.017), not in the pMTG or IFG (ps > 0.58). Main effects of group were not found (ps > 0.18).

We performed a whole-brain analysis of the semantic abstractness effects in the two deaf groups (*Figure 5—figure supplement 1*). For native signers, at the threshold of voxel-level p < 0.001, cluster-level FWE-corrected p < 0.05, semantic abstractness reached significance in the left dATL (peak MNI coordinates: –58, 4,–8, peak t = 7.27, 209 voxels). For delayed signers, no regions survived the whole-brain threshold; at a lenient threshold of voxel p < 0.001, cluster size >10 voxels, abstractness could be observed in the left pMTG and dATL, which are consistent with the ROI results above. The whole-brain between-group comparisons of the semantic abstractness effects did not yield any areas surviving the conventional whole-brain correction threshold.

## Controlling for the written language performances for dATL semantic reduction in delayed signers

Delayed L1 acquisition may associate with language proficiency changes in multiple aspects (see *Table 1*). It is possible that the dATL semantic computation is not associated with delayed L1 acquisition per se, but with subjects' written language performance, a key source of semantic knowledge accumulation. Our two groups of deaf signers did not significantly differ in their performances in two word-reading tasks (*Figure 1—figure supplement 1*), which was consistent with the self-rated reading comprehension ability in the ARHQ (Welch's $t_{35.6}$ = –0.90, p = 0.375, Hedges' *g* = –0.28, *Table 1*). However, delayed signers did report significantly lower writing ability (Welch's $t_{31.0}$ = –2.91, p = 0.007, Hedges' *g* = –0.93) and a tendency of lower reading speeds (Welch's $t_{26.2}$ = –1.89, p = 0.07, Hedges' *g* = –0.62) in the ARHQ than native signers (*Table 1*). We then carried out mediation analyses (using the *mediation* package in R) to examine whether writing or reading speed mediated the group differences in neural semantic effects in the left dATL. For both the multivariate RSA and univariate abstractness effects, the direct effects of group remained largely significant (ps < 0.074), whereas the mediation effects did not approach statistical significance (ps > 0.52), based on 5000 times bootstrapping resampling. That is, the reduced dATL semantic effects induced by delayed L1 acquisition were not fully attributable to written language processes.

## Discussion

By comparing fMRI BOLD responses to the word semantic structures between native and delayed early deaf signers, we aimed to examine how early-life language exposure affects the semantic neural representations. Our results demonstrated that semantic information in the left dATL was significantly reduced in delayed signers compared with native signers in both multivariate (representation of rich semantic space) and univariate (preference to abstract/nonobject words) analyses. That is, early-life language acquisition – critical to language system neurodevelopment – is necessary for the left dATL to exhibit the typical semantic organization in adulthood. These results provide the first positive evidence for the effect of language experience on driving the semantic functional development of a particular brain region.

Two types of specificity are to be clarified – anatomical specificity and information specificity. First, is this group effect specific to the dATL, relative to other brain regions? We do not have evidence for such a strong region specificity. We did not observe a significant group-by-ROI interaction in either the RSA or the univariate analyses. That is, the group differences were not significantly stronger in the dATL than in the other semantic regions being analyzed (IFG and pMTG). We are thus not claiming that the dATL is the only region that derives semantic representation from language experience, but choose to focus the following discussion on this region because of the robust positive effects here. Second, is the dATL semantic representation specifically derived from language experience and not other (nonlinguistic) sensory experiences? The manipulation of the current study – the two groups of deaf signers – varies on the language exposure while matching on the sensory experiences (see below), and thus did not test the presence or absence of sensory-derived semantic representations.

Inferences could be drawn in combination with the previous studies that focused on the manipulation of sensory experiences by studying visual knowledge in congenitally blind subjects. There, it was reported that the blind and sighted had comparable semantic information encoding in the RSA (*Wang et al., 2020*). In terms of univariate effects (abstractness or color concept adaptation), deprivation of sensory experiences did not reduce, but actually tended to enhance the effects here (*Bottini et al., 2020*; *Striem-Amit et al., 2018*). That is, evidence from congenitally blind studies does not support the additional sensory-derived semantic encoding here. While drawing negative conclusions is always difficult, we reason that it is parsimonious, based on available data, to propose that the dATL's contribution to semantic encoding is specific to language, and not sensory-derived representations.

What kind of variables best account for the observed group differences in the dATL? The two deaf groups were matched on sensory experiences (both similarly profoundly deaf), a wide range of nonlinguistic environmental variables including the socioeconomic status (except for parents' education backgrounds, which were better for the delayed signer group), the length and level of formal education, through which largely comparable sign language and word reading skills were achieved. The salient difference was the early-life language experience (before 6.9 years of age – the mean age of acquisition in our delayed signer group). Notably, deaf children with limited access to natural language often spontaneously develop homesign systems, which share some aspects with natural language (*Goldin-Meadow, 2003*). For instance, they could use homesigns to produce generic utterances in a similar manner to typically developed children (*Goldin-Meadow et al., 2005*). Notably, routine nation-wide neonate hearing screening in China did not start until 2009, years after the early childhood of our participants (born before 2000), and some hearing parents may nonetheless try to give deaf children additional aids of exposure to signs (via preschool special education programs) or speech (via hearing aids). Critically, our positive results of the robust group differences in the dATL suggest that early homesign/aid measures and later formal education for sign and written language experiences are insufficient for typical dATL neurodevelopment; the full-fledged language experience during early infancy and childhood (before school age) plays a necessary role in this process. A further variable to consider is that a lack of early language exposure may lead to alterations beyond the language system, affecting non-language cognitive domains such as working memory or theory of mind in children (*Marshall et al., 2015*; *Richardson et al., 2020*). While we cannot rule out the effects of these potential intermediate cognitive processes, the positive evidence of these variables in dATL functionality is lacking. The most direct manipulation and parsimonious account for the dATL effects is language-related.

The effects of language experience in shaping the semantic representation in the dATL observed here are corroborated by several previous lines of indirect evidence for the relevance of language in its functionality. It shows a functional preference for abstract words, which are less salient in sensory attributes than concrete words and are assumed to entail more language processes such as context diversity (e.g. *Binder et al., 2009*; *Bucur and Papagno, 2021*; *Wang et al., 2010*). In congenitally blind individuals, it encodes visual knowledge such as color and the activity strength is negatively modulated by perceptibility (*Bottini et al., 2020*; *Striem-Amit et al., 2018*; *Wang et al., 2020*). It is sensitive to semantic composition, such as in two-word phrases or sentences (*Pallier et al., 2011*; *Pylkkänen, 2019*). This region is functionally and structurally connected with other language-sensitive regions (*Fan et al., 2014*; *Friederici et al., 2017*; *Ralph et al., 2017*; *Pascual et al., 2015*; *Wang et al., 2019*). Finally, a recent neurodevelopmental study shows that the early-life resting-state functional connectivity patterns of the left temporal pole significantly predict performances in tasks entailing semantic processes, not in tasks of phonological skills or rapid automatized naming, at about 6.5 years of age (*Yu et al., 2021*). These lines of findings are often correlational evidence for the language effect on their own. Together with the positive evidence of early language experience in developing typical semantic representation in the dATL reported here, the parsimonious proposal accounting for them together would be that the dATL represents knowledge derived from language (e.g. from higher-order word relations). Without early language exposure, the semantic structure representation and the abstractness preference here are reduced.

Together, this package of findings extends beyond the majority of the neurocognitive semantic theories, which mainly focus on the representing, binding, and controlling of knowledge derived from nonlinguistic multisensory experiences (*Binder, 2016*; *Ralph et al., 2017*; *Martin, 2016*; but see *Mahon and Caramazza, 2008*). The recently developed dual-neural-coding framework for semantic

knowledge (*Bi, 2021*), which explicitly proposes that the human brain has developed regions (the dATL) for language-derived semantic representations, in addition to the sensory-derived knowledge representations, naturally accommodates these observations. Language, as a symbolic system, allows us to convey and manipulate meanings free from sensorimotor experiences and plays a key role in knowledge transmission and accumulation across individuals and generations (*Gelman and Roberts, 2017*). The existence of the language-derived knowledge neural system may lay the foundation for the efficient storage and manipulation of semantic knowledge, particularly abstract knowledge without tangible and consistent sensorimotor referents (*Bi, 2021*).

The current results also have implications for the role of early language exposure in neurodevelopment more generally. Previous studies have reported that delayed L1 acquisition associates with reduced cortical volume in the left inferior frontal region, reduced cortical thickness in the left posterior middle temporal region, and the reduced fractional anisotropy values in the left arcuate fasciculus that structurally connects the two regions, and not ATL or its related white matter tracts (*Cheng et al., 2023*; *Cheng et al., 2019*). Our findings of semantic functional alterations in the dATL are thus not easily attributable to broad anatomical changes associated with late L1 acquisition. Notably, different from phonological and syntactic processes, where both visible behavioral underdevelopment (e.g. *Caselli et al., 2021*; *Cheng and Mayberry, 2021*; *Mayberry et al., 2002*) and brain functional changes (*Mayberry et al., 2011*; *Richardson et al., 2020*; *Twomey et al., 2020*) were observed, for semantics we only observed brain functional changes in the dATL but no visible behavioral effects. Consistent with the literature where deaf delayed signers did not show differences to controls in semantic interference effects in the picture-sign paradigm (*Baus et al., 2008*), scalar implicature (*Davidson and Mayberry, 2015*), or N400 measures (*Skotara et al., 2012*), we did not observe visible differences in terms of semantic distance structures (*Figure 1a*) or reaction time of lexical decision and word-triplet semantic judgment (*Figure 1—figure supplement 1*). As reasoned in the Introduction, this seeming neuro-behavior discrepancy might be related to the multifaceted, distributed nature of the cognitive and neural basis of semantics more broadly. The general semantic behavioral tasks we employed could be achieved with representations derived from multiple types of experiences, supported by highly distributed neural systems (e.g. *Bi, 2021*; *Binder and Desai, 2011*; *Ralph et al., 2017*; *Martin, 2016*), including those not affected by delayed L1 acquisition in regions beyond the dATL. This finding invites future studies to specify the exact developmental mechanisms in the left dATL (*Fu et al., 2023*; *Unger and Fisher, 2021*) and to uncover semantic behavioral consequences related to the functionality of this area.

To conclude, by manipulation of early language experience by nature we were able to positively identify a human brain region – dATL – robustly supporting word meanings derived from language experience, in contrast to those grounding nonlinguistic, sensory experiences. These findings highlight the role of language in forming specific neural semantic representations in the human brain, and raise new questions about the ontogenetic mechanisms of this intriguing neural structure.

## Materials and methods

### Participants

Thirty-nine congenitally or early deaf adults and 33 hearing college students (15 males, mean age 21.97 ± 2.58 years, range: 18–28 years, all native Mandarin Chinese speakers) were recruited for the fMRI study. All except for one delayed deaf signer and one hearing participant participated in the behavioral semantic distance judgment task. The behavioral data of one delayed signer were excluded from data analysis due to technical errors. The behavioral and neural data of 21 out of the 33 hearing college students overlapped with a previous study (*Wang and Bi, 2021*) that examined the intersubject variability of semantics in typically developed populations.

The deaf participants included two groups (*Table 1*). Native signers (n = 16; 11 males) had deaf parents and were exposed to CSL soon after birth. Delayed (nonnative) signers (n = 23; 12 males) were born in hearing families and learned CSL after they were enrolled in special education schools (range: 4–10 years of age). All deaf participants completed a background questionnaire, in which they reported their hearing loss conditions, history of language acquisition, and educational background. They were also asked to rate sign language abilities of their family members and to rate their own sign and lipreading abilities using a 7-point scale: 1 = none at all, 7 = very proficient. All deaf

participants then finished the Chinese version of the ARHQ (a self-report screening tool for the risk of reading disability in adults, *Lefly and Pennington, 2000*), the MacArthur Scale of subjective social status (*Adler et al., 2000*) during childhood, and reported the educational years of their parents. For hearing loss conditions, all deaf participants reported being severely or profoundly deaf from birth, except for one native signer and three delayed signers who reported becoming deaf before the age of 2 due to medication side effects. Hearing thresholds in decibels (dB) were available in 23 deaf participants (native: 11/16; delayed: 12/23) and confirmed severe to profound hearing loss (range: 85–120 dB). One native signer and four delayed signers were using hearing aids at the time of testing; others either had never used hearing aids (six native signers and five delayed signers) or had used hearing aids for some time (nine native signers and fourteen delayed signers; years of use: 0.5–20 years). Speech comprehension was reported to be very poor, even with the use of hearing aids.

All participants had normal or corrected-to-normal vision. All participants were right-handed, except for three native signers (one was ambidextrous and two were left-handed) (measured by the Edinburgh inventory, *Oldfield, 1971*). All participants gave written informed consent and received monetary compensation for participation. The study was approved by the Human Subject Review Committee at Peking University (2017-09-01), China, in accordance with the Declaration of Helsinki.

## Word stimuli in task fMRI

The stimuli for fMRI scanning were 90 written words, including 40 concrete/object words and 50 abstract/nonobject words without explicit external referents. Concrete/object words varied in sensory and motor attributes and included 10 animals (e.g. *panda*), 10 face or body parts (e.g. *shoulder*), 10 tools (e.g. *hammer*), and 10 common large household objects (e.g. *microwave*). Abstract/nonobject words varied in social and emotional associations (e.g. *cause*, *violence*). A 7-point familiarity rating, with 7 being the most familiar, was collected from 21 out of 39 deaf signers and an independent group of 26 hearing college students. All words were highly familiar to deaf participants (8 native signers: 6.72 ± 0.17; 13 delayed signers: 6.86 ± 0.15). All words were disyllabic except for five object words ('cat' and 'bed' are monosyllabic and 'giraffe', 'microwave', and 'washing machine' are trisyllabic in Chinese). All words were primarily used as nouns except for 10 words denoting emotional states (nine were primarily used as adjectives and one as a verb). The concrete and abstract words were matched on the number of strokes (a measure of visual complexity for Chinese words; 17.23 ± 5.80 vs 16.14 ± 3.98; $t_{88} = 1.05$, p = 0.30). While concrete/object words were less frequent in a Mandarin Chinese corpus (*Sun et al., 1997*) than abstract/nonobject words (log word frequency: 1.05±0.73 vs 1.62 ± 0.68; $t_{88} = -3.83$, p < 0.001), the former were rated as more subjectively familiar than the latter in either hearing (6.79 ± 0.16 vs 6.22 ± 0.26; $t_{88} = 12.05$, p < 0.001) or deaf (6.85 ± 0.07 vs 6.77 ± 0.13; $t_{88} = 3.41$, p < 0.001) participants.

## Behavioral semantic distance judgment task

To examine participants' understanding of the 90 words used in the task fMRI, we asked them to judge semantic distance among these words by arranging them spatially close together or far apart in a circular arena on a computer screen via mouse drag-and-drop operations (*Kriegeskorte and Mur, 2012*). The task lasted for 1 hr and produced a 90 × 90 semantic distance matrix for each participant. We correlated each deaf participant's semantic RDM with the hearing group-averaged semantic RDM using Spearman's rank correlation, Fisher-*z*-transformed the correlation coefficients, and compared the two deaf groups to assess the similarity between their semantic structures.

Considering the multidimensional and flexible nature of semantic distance among various concrete and abstract words (*Binder et al., 2016*; *Conca et al., 2021*), we focused on the categorical structure of 90 words to boost signal and to allow for better generalization across items (i.e. along the categorical structure). The categorical RDM was obtained by performing a k-means clustering analysis on the group-averaged 90 × 90 semantic RDM of hearing participants (the factoextra package, http://www.sthda.com/english/rpkgs/factoextra, RRID: SCR_016692; in the R programming environment, version 4.0.0; *R Development Core Team, 2020*). The optimal number of clusters was determined based on gap statistic (*Tibshirani et al., 2001*), which revealed 10 semantic categories (*Figure 1a*).

## Task fMRI procedure

In the MRI scanner, participants were instructed to look at each of the 90 target words, think about their meanings, and perform an oddball one-back semantic judgment task. In this oddball task, participants were asked to judge whether occasional words in red (catch trials) were semantically related to the previous word by pressing the corresponding buttons with the right index or middle finger.

Each participant performed 10 runs (360 s per run) of task fMRI scanning, except for one native signer who finished eight runs and withdrew due to discomfort. Each run consisted of 90 2.5-s-long target word trials and 14 2.5-s-long catch trials. Each trial started with a 0.8 s word stimulus (black color, SONG font, 2.6 visual degrees in height) at the center of a gray background, followed by 1.7 s fixation. Thirty 2.5-s-long null trials (fixation only) were randomly inserted among target and catch trials, with the interval between two words ranging from 1.7 to 11.7 s. Each target word appeared once in each run, and the order was pseudo-randomized for each run in each participant. Each run began with a 12 s fixation period and ended with a 13 s rest period during which participants received a written cue that the current run was about to end. Stimulus presentation was controlled by E-prime 2 (Psychology Software Tools, Inc, Pittsburgh, PA, USA). The 140 catch trials were created by first pairing each of the 90 target words with a probe word and then pseudo-randomly selecting 50 out of 90 target words (21 concrete/object words and 29 abstract/nonobject words) to pair with another 50 probe words. Participants performed this task attentively, with overall miss rates lower than 22.1% (median: 1.4%).

## Image acquisition

All functional and structural MRI data were collected using a Siemens Prisma 3T Scanner with a 64-channel head-neck coil at the Center for MRI Research, Peking University. Functional data were acquired with a simultaneous multi-slice echoplanar imaging sequence supplied by Siemens (62 axial slices, repetition time [TR]=2000 ms, echo time [TE]=30 ms, multi-band factor = 2, flip angle [FA]=90°, field of view [FOV]=224 mm × 224 mm, matrix size = 112 × 112, slice thickness = 2 mm, gap = 0.2 mm, and voxel size = 2 mm × 2 mm × 2.2 mm). A high-resolution 3D T1-weighted anatomical scan was acquired using the magnetization-prepared rapid acquisition gradient echo sequence (192 sagittal slices, TR = 2530 ms, TE = 2.98 ms, inversion time = 1100 ms, FA = 7°, FOV = 224 mm × 256 mm, matrix size = 224 × 256, interpolated to 448 × 512, slice thickness = 1 mm, and voxel size = 0.5 mm × 0.5 mm × 1 mm).

## Image preprocessing

Functional images were preprocessed using SPM12 (http://www.fil.ion.ucl.ac.uk/spm12/; RRID: SCR_007037). For each participant, the first four volumes of each functional run were discarded for signal equilibrium. The remaining images were corrected for slice timing and head motion and then spatially normalized to Montreal Neurological Institute (MNI) space via unified segmentation (resampling into 2 mm × 2 mm × 2 mm voxel size). All participants had head motion less than 1.97 mm/1.95°, except for one hearing participant showing excessive head motion in 2 runs (>2 mm/2°); we thus analyzed the remaining 8 runs of fMRI data for this participant. We also estimated each participant's framewise displacement (FD) from translations and rotations (*Power et al., 2012*); the two deaf groups exhibited comparable FD during scanning (native: 0.12±0.04; delayed: 0.11±0.04; Welch's $t_{29.1}$ = 0.77, p = 0.45). Spatial smoothing was applied with a Gaussian kernel of 6 mm full width at half maximum (FWHM) for univariate analysis and 2 mm FWHM for RSA.

## ROI definition

A GLM was built to localize semantic abstractness in the brain, that is, brain regions showing stronger activations to abstract/nonobject words than to concrete/object words, in hearing participants. These regions were considered as candidates supporting knowledge derived from language. For spatially smoothed functional images, the GLM included three regressors (i.e. abstract/nonobject words, concrete/object words, and catch trials) for each run, each convolved with the canonical hemodynamic response function (HRF). The GLM also included six head motion parameters and a global mean predictor of each run. The high-pass filter was set at 128 s. The contrast (i.e. abstract > concrete) was computed for each participant.

The beta-weight images of abstract > concrete in hearing participants were submitted to one-sample *t*-tests. A conventional cluster-extent-based inference threshold (voxel-level p < 0.001, cluster-level FWE-corrected p < 0.05) was adopted and we stated explicitly when other thresholds were applied. At the conventional threshold, six clusters were found (*Figure 1c*). As the largest cluster extended from the left dATL to the left IFG, we separated the dATL (808 voxels) and IFG (411 voxels) into two ROIs based on the automated anatomical labeling (AAL) atlas (*Tzourio-Mazoyer et al., 2002*). To have the cluster sizes comparable across ROIs, we increased voxel-level threshold to p < 0.0001 and obtained a smaller dATL cluster (488 voxels), based on which the ROI-level results were reported; the results were largely similar across different sizes of the dATL ROI.

## Representational similarity analysis
### GLM
The preprocessed functional images were analyzed using a GLM to create a *t*-statistic image for each word (*Kriegeskorte et al., 2008*). For each participant, a GLM was built with the concatenated time series across all the scanning runs, including 90 regressors corresponding to each word and one regressor for catch trials, convolved with a canonical HRF. Additionally, six head motion parameters and a global mean predictor were included for each run. A high-pass filter cutoff was set as 128 s. The resulting *t*-maps for each word versus baseline were used to create neural RDMs.

### ROI-level RSA
For each ROI in each participant, we extracted the activity patterns to each word from its whole-brain *t*-statistic images and calculated a neural 90 × 90 RDM based on the Pearson distance between activation patterns for each pair of words. Semantic information encoded in this ROI was quantified by computing Spearman's rank correlation between the neural RDM and the 10-category benchmark semantic RDM. Partial correlations were used to control for two stimulus low-level property RDMs: (1) The low-level visual similarity RDM was computed by Pearson's correlation distance between the binary silhouette images of each word pair. (2) The word phonological RDM: Despite that our task did not require explicit phonological retrieval, there might be automatic phonological activation. We thus constructed the phonological RDM by calculating one minus the proportion of shared sub-syllabic units (onsets or rhyme) between each word pair. The resulting correlation coefficients between neural and semantic RDMs were Fisher-*z*-transformed and compared with zero using one-sample *t*-tests (one-tailed) to test whether the ROI significantly encoded the semantic structure. Group differences in semantic encoding were assessed using two-tailed Welch's *t*-tests for each ROI.

### Whole-brain searchlight RSA
To explore the potential differences between native and delayed signers in the neural semantic representations beyond the semantic ROIs identified above, we carried out whole-brain searchlight RSA. For each voxel in the AAL mask, we extracted the multivoxel activity patterns of 90 words within a sphere (radius = 8 mm) centered at that voxel. A neural 90 × 90 RDM was computed based on Pearson distance and was correlated with the semantic benchmark RDM while controlling for the two low-level RDMs of visual and phonological similarity of word stimuli, which produced a correlation coefficient for this voxel. By moving the searchlight center through the AAL mask, we obtained a correlation map for each participant. This map was Fisher-*z*-transformed and then spatially smoothed using a 6 mm FWHM Gaussian kernel. The correlation maps of native and delayed groups were then compared using a two-sample *t*-test.

## Univariate semantic abstractness analysis
### GLM
The neural semantic abstractness effects were computed using the same GLMs in the 'ROI definition' section. For each deaf participant, we computed the following three contrasts for ROI and whole-brain group comparisons using all the runs: abstract > concrete, abstract words > baseline, concrete words > baseline.

### ROI analysis
For each ROI, we extracted the averaged beta values of abstract words > baseline and concrete words > baseline, from each deaf participant, and compared these beta values of the two word types

using one-tailed paired *t*-tests to test for the semantic abstractness effects in each deaf group. Two-way ANOVA was then adopted to examine group effects, with word type (concrete/object, abstract/nonobject) as a within-subject factor and group (native, delayed) as a between-subject factor.

### fROI analysis

We also carried out individualized fROI analyses (*Cohen et al., 2019*; *Ratan Murty et al., 2020*), which examined semantic selectivity in individually defined functional voxels, with the fROI localization and selectivity calculation using independent datasets. Specifically, we estimated each deaf participant's whole-brain beta and *t* maps for the contrast between abstract/nonobject and concrete/object words in odd and even runs. The semantic ROIs defined above in hearing participants were taken as anatomical constraints. In each hearing-group-ROI, for each deaf participant, we localized his/her top 50 selective voxels (i.e. voxels with the highest *t* values to the contrast) in the odd runs, extracted the mean beta values of these voxels to abstract and concrete words, respectively, in the even runs. This procedure was repeated with fROI defined in the even runs and beta values calculated in the odd runs. The beta values were averaged across two iterations for each ROI in each participant and compared between the two deaf groups using the abovementioned statistical analyses. We also repeated the fROI analyses at the fROI size of 100 voxels and obtained very similar results.

### Whole-brain analysis

The whole-brain semantic abstractness effects were examined by one-sample *t*-tests on the whole-brain beta-weight images of abstract > concrete in each group. For the whole-brain group comparison of the semantic abstractness effects, the abstract > concrete beta-weight maps of the two deaf groups were submitted to a two-sample *t*-test.

## Brain visualization

The brain results were projected onto the MNI brain surface for visualization using BrainNet Viewer (version 1.7; https://www.nitrc.org/projects/bnv/; RRID: SCR_009446; *Xia et al., 2013*) with the default 'interpolated' mapping algorithm, unless stated explicitly otherwise. Regional labels were derived based on the AAL template in xjview (by Xu Cui, http://www.alivelearn.net/xjview/; RRID: SCR_008642).

## Acknowledgements

We thank Dr. Xi Yu for helpful discussions on earlier drafts of the manuscript. We thank Ms. Yun Hao, Anran Deng, and Wei Liang for their assistance in data collection. This work was supported by STI2030-Major Project 2021ZD0204100 (2021ZD0204104), the National Natural Science Foundation of China (31925020 and 82021004 to YB, 32171052 and 31700943 to XW), and Changjiang Scholar Professorship Award (T2016031 to YB). The funders had no role in study design, data collection, and interpretation, or the decision to submit the work for publication.

## Additional information

### Competing interests

Yanchao Bi: Senior Editor, eLife. The other authors declare that no competing interests exist.

### Funding

| Funder | Grant reference number | Author |
| --- | --- | --- |
| STI2030-Major Project 2021ZD0204100 | 2021ZD0204104 | Yanchao Bi |
| National Natural Science Foundation of China | 31925020 | Yanchao Bi |
| National Natural Science Foundation of China | 82021004 | Yanchao Bi |

| Funder | Grant reference number | Author |
|--------|------------------------|--------|
| National Natural Science Foundation of China | 32171052 | Xiaosha Wang |
| National Natural Science Foundation of China | 31700943 | Xiaosha Wang |
| Changjiang Scholar Program of Chinese Ministry of Education | Professorship Award T2016031 | Yanchao Bi |

The funders had no role in study design, data collection and interpretation, or the decision to submit the work for publication.

### Author contributions
Xiaosha Wang, Conceptualization, Data curation, Software, Formal analysis, Funding acquisition, Validation, Investigation, Visualization, Methodology, Writing – original draft, Project administration, Writing – review and editing; Bijun Wang, Data curation, Formal analysis, Methodology, Project administration; Yanchao Bi, Conceptualization, Resources, Supervision, Funding acquisition, Investigation, Writing – original draft, Project administration, Writing – review and editing

### Author ORCIDs
Xiaosha Wang  http://orcid.org/0000-0002-2133-8161
Yanchao Bi  http://orcid.org/0000-0002-0522-3372

### Ethics
Human subjects: All participants gave written informed consent and received monetary compensation for participation. The study was approved by the Human Subject Review Committee at Peking University (2017-09-01), China, in accordance with the Declaration of Helsinki.

### Decision letter and Author response
Decision letter https://doi.org/10.7554/eLife.81681.sa1
Author response https://doi.org/10.7554/eLife.81681.sa2

## Additional files

### Supplementary files
• MDAR checklist

### Data availability
Source data files have been provided for Table 1 and all the figures. Additional behavioral and neural data have been made available on OSF at the link https://osf.io/wz6q9/. The whole-brain unthresholded statistical maps have also been made available on NeuroVault at the link https://neurovault.org/collections/13705/. Deidentified Nifti files are not shared openly because of ethics constraints, but are available from the corresponding authors upon reasonable request.

The following datasets were generated:

| Author(s) | Year | Dataset title | Dataset URL | Database and Identifier |
|-----------|------|---------------|-------------|-------------------------|
| Wang X | 2023 | Early language exposure affects neural mechanisms of semantic representations | https:/doi.org/10.17605/OSF.IO/WZ6Q9 | Open Science Framework, 10.17605/OSF.IO/WZ6Q9 |
| Wang X | 2023 | Early language exposure affects neural mechanisms of semantic representations | https://neurovault.org/collections/13705/ | NeuroVault, 13705 |

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
