## [Editor Report]

This study provides important evidence regarding the development of concept representations, using functional brain imaging to compare concept structure in people with different amounts of language experience. The analyses, which are overall solid, suggest that concept representations differ as a function of childhood language experience.

---

## [Decision Letter]

**Decision letter after peer review:**

Thank you for submitting your article "Early-life language deprivation affects specific neural mechanisms of semantic representations" for consideration by *eLife*. Your article has been reviewed by 3 peer reviewers, one of whom is a member of our Board of Reviewing Editors, and the evaluation has been overseen by Floris de Lange as the Senior Editor. The following individuals involved in the review of your submission have agreed to reveal their identity: Jamie Reilly (Reviewer #2); Scott Laurence Fairhall (Reviewer #3).

Essential revisions:

1) Expanding on the discussion of critical periods for language development, particularly with respect to lexical-semantic and semantic acquisition (as opposed to phonology and syntax).

2) Explicitly evaluating whether the dATL differs in pattern from other ROIs, which would be necessary to say strongly that the dATL is "specifically" or "uniquely" sensitive.

3) Addressing what seems to be a mismatch between the behavioral results (no group differences) and brain results (significant group differences).

4) Before resubmitting please revisit the language regarding the language-deprived group carefully to ensure the logic is clear and not stigmatizing. There was some disagreement among reviewers during a discussion over this issue so I just ask you to ensure the issue is handled thoughtfully.

*Reviewer #1 (Recommendations for the authors):*

– Please share the deidentified Nifti files on OpenNeuro, or if this is not permitted by ethics constraints, include the reason in the manuscript.

– Please consider sharing results maps on NeuroVault rather than OSF to aid discoverability (and NeuroVault comes with a viewer that facilitates viewing unthresholded maps).

– Many of the paragraphs are quite long (spanning a page or more). I would suggest considering breaking some of these into smaller paragraphs, with informative topic sentences, to enhance readability.

– Without cytoarchitectonic information, including the Brodmann area may not be well justified. A helpful discussion on this can be found in Devlin and Poldrack (2007), In praise of tedious anatomy.

*Reviewer #2 (Recommendations for the authors):*

My critique was not focused on the viability data but more on how to better frame assumptions and predictions. The data are very promising and interesting. I felt that a clear set of front-ended predictions would be very helpful, as would elaborating on the role of language deprivation.

*Reviewer #3 (Recommendations for the authors):*

The study is well-written, the work is well-motivated and the analysis is consistent with the field. I have a few specific comments below.

Comment 1

The authors make some strong claims and it may be worth being a little more precise about exactly what is being claimed. For instance, if we break down the authors concluding sentence from the abstract "These results provide positive, causal evidence that the neural semantic representation in dATL is specifically supported by language, as a unique mechanism of representing (abstract) semantic space, beyond the sensory-derived semantic representations distributed in the other cortical regions."

By 'specifically', do the authors mean that dATL is a solely language-based representation (and sensory-derived conceptual representation plays no role, even in the sighted), do they mean the dATL specifically (and not other brain regions) is involved in language-derived conceptual representation. Does 'supported', mean a supportive/auxiliary role, or does this mean that language is the basis upon which representations are formed? Does 'unique' imply that the dATL alone is language-derived, that it is alone in being language derived and independent of sensory-derived representations, that is the unique representation of truly abstract knowledge?

I would encourage the authors to, as much as possible, clearly lay out their claims, as this is of genuine interest (rather than just 'softening the language').

Comment 2

I do get the sense that the authors are proposing that dATL may be purely language-derived representation not grounded in or in any derived from sensory experience. It seems possible that this can be the case, as shown in the congenitally blind population, but this does not indicate that it is always the case. It is very possible that in the absence of sensory experience, dATL representations can continue to form, but it is simultaneously possible that when this sensory experience is present, more grounded processes also contribute. Are the authors arguing that the present result shows that this is not the case? Please clarify.

Comment 3

The main empirical emphasis of this work and the authors' general thesis is the specificity or particular 'peak' importance of the dATL. The group difference in the RSA and in the univariate analysis of the abstract>concrete effect in dATL do indeed support the importance of dATL. However, the failure of these effects to reach significance in the other abstract>concrete ROIs (pMTG, IFG) does not indicate that these effects are different between dATL and (e.g.) pMTG.

Ideally, ROI by group interactions would be performed on the ROI data in Figure 2B and Figure 3A to show that dATL is statistically different from the other ROIs. I appreciate that this level of statistical vigour is often not met in our field but considering the importance and interesting nature of the authors' claims, I feel it would strongly improve the quality of the results here.

If the ROI by effect interactions are not significant, then a clear caution about over-interpreting the absence of effects of pMTG and IFG could be included in the discussion.

Comment 4

Similarly, the searchlight analysis (lines 220-229) reports the presence of effects in native signers and the absence of effects in later signers (while there are no significant differences between the groups). This may also encourage the reader to over-interpret the absence of effects. The inclusion of the searchlight is clearly motivated, in that it shows that group differences are not apparent outside the ROIs analysed in the previous section. It would be good if the authors could emphasise more strongly the absence of effects outside of the ROIs and caution against the over-interpretation of the absence of effects in delayed signers given the absence of significant differences between delayed and native signers.

Comment 5

Relating to comment 4, if we are to take the descriptive route, it seems there is comparable evidence implicating the left AG/TPJ in language-derived representations in native but not delayed signers (Figure 3C). Might a (post hoc) ROI analysis indicate comparable effects of delayed language acquisition in this region? How would the potential generalisability of this effect impact the authors' beliefs about dATL?

Comment 6

Delayed language acquisition could have potentially wide-reaching effects on cortical development. I appreciate that participants were matched on some criteria (e.g. self-report sign proficiency, lexical decision) but do the authors have reason to be sure that reported imaging differences are not a non-specific result of broad cortical changes? One supportive data point I can think of is the difference between dATL and IFG but perhaps there is information in the literature. It would be good if the authors could lay these out.

Comment 7

Do the authors think that language-derived knowledge is related to all domains dATL? For instance, in congenitally blind individuals, do the authors think distinctions between a mountain range versus a city skyline would be present in dATL rather than PPA or RSA?

---

## [Author Response]

Essential revisions:1) Expanding on the discussion of critical periods for language development, particularly with respect to lexical-semantic and semantic acquisition (as opposed to phonology and syntax).

We appreciate the suggestion and have revised our manuscript thoroughly in light of this comment. First, we now avoid the vague term “critical period”, which may be taken to refer to critical period for different, specific cognitive and/or neural development in the literature. Instead, we now made explicit throughout the specific processes being discussed (phonology, syntax, semantics). The behavioral and neural effects of early language experience (delayed language acquisition) on phonology, syntax, and semantics are now elaborated, discussed separately and explicitly in both the Introduction and Discussion (see more detailed responses below, Reviewer 2 – Comment 1).

2) Explicitly evaluating whether the dATL differs in pattern from other ROIs, which would be necessary to say strongly that the dATL is "specifically" or "uniquely" sensitive.

We added two-way ANOVA analyses, with ROIs as the within-subject factor and group as the between-subject factor. The ROI-by-group interaction did not reach significance for either the RSA semantic encoding effect or the univariate abstractness effect (pages 9 and 12). We have revised the statistical reports and discussions accordingly, adding a paragraph explicitly stating that there is no evidence for a strong anatomical specificity (Discussion paragraph 2, page 14). Please see also below (Reviewer 3 – Comment 3) for more detailed responses.

3) Addressing what seems to be a mismatch between the behavioral results (no group differences) and brain results (significant group differences).

We have added explicit discussions addressing this issue (page 18):

“Consistent with the literature where deaf delayed signers did not show differences to controls in semantic interference effects in the picture-sign paradigm (Baus et al., 2008), scalar implicature (Davidson and Mayberry, 2015) or N400 measures (Skotara et al., 2012), we did not observe visible differences in terms of semantic distance structures (Figure 1a) or reaction time of lexical decision and word-triplet semantic judgment (Supplementary file 1). As reasoned in the Introduction, this seeming neuro-behavior discrepancy might be related to the multifaceted, distributed nature of the cognitive and neural basis of semantics more broadly. The general semantic behavioral tasks we employed could be achieved with representations derived from multiple types of experiences, supported by highly distributed neural systems (e.g., Bi, 2021; Binder and Desai, 2011; Lambon Ralph et al., 2017; Martin, 2016), including those not affected by delayed L1 acquisition in regions beyond dATL. This finding invites future studies to specify the exact developmental mechanisms in the left dATL (Fu et al., 2022; Unger and Fisher, 2021) and to uncover semantic behavioral consequences related to the functionality of this area.”

Also in the Introduction, the potential differences between behavioral and neural profiles for semantics are laid out, motivating the importance of looking at semantic neural representations (Intro paragraph 4, page 4).

4) Before resubmitting please revisit the language regarding the language-deprived group carefully to ensure the logic is clear and not stigmatizing. There was some disagreement among reviewers during a discussion over this issue so I just ask you to ensure the issue is handled thoughtfully.

Thank you for this kind reminder. With references to related studies in the literature, we removed the term “language deprivation” and used the terms “subjects with varying amounts of early language exposure” or “delayed L1 acquisition” to more precisely describe our experimental manipulation throughout the revised manuscript.

Reviewer #1 (Recommendations for the authors):– Please share the deidentified Nifti files on OpenNeuro, or if this is not permitted by ethics constraints, include the reason in the manuscript.

Added in the manuscript “Deidentified Nifti files are not shared openly because of ethics constraints, but are available from the corresponding authors upon reasonable request.” (page 28)

– Please consider sharing results maps on NeuroVault rather than OSF to aid discoverability (and NeuroVault comes with a viewer that facilitates viewing unthresholded maps).

Done. “The whole-brain unthresholded statistical maps have also been made available on NeuroVault at the link https://neurovault.org/collections/13705/.” (page 28)

– Many of the paragraphs are quite long (spanning a page or more). I would suggest considering breaking some of these into smaller paragraphs, with informative topic sentences, to enhance readability.

Done.

– Without cytoarchitectonic information, including the Brodmann area may not be well justified. A helpful discussion on this can be found in Devlin and Poldrack (2007), In praise of tedious anatomy.

Thank you. We have removed the information of Brodmann areas from the Results section.

Reviewer #2 (Recommendations for the authors):My critique was not focused on the viability data but more on how to better frame assumptions and predictions. The data are very promising and interesting. I felt that a clear set of front-ended predictions would be very helpful, as would elaborating on the role of language deprivation.

Thank you for your constructive comments on our study. Following your and other reviewers’ suggestions, we have thoroughly revised the Introduction section. The major revisions include: (1) to elaborate on the previous findings of delayed L1 acquisition on the behavioral and neural effects of semantic knowledge; (2) to be more explicit about our focus and predictions on the effects of delayed L1 acquisition on the neural representations of semantics (pages 2, 4-5).

Reviewer #3 (Recommendations for the authors):The study is well-written, the work is well-motivated and the analysis is consistent with the field. I have a few specific comments below.

We appreciate the positive evaluations and the constructive comments.

Comment 1The authors make some strong claims and it may be worth being a little more precise about exactly what is being claimed. For instance, if we break down the authors concluding sentence from the abstract "These results provide positive, causal evidence that the neural semantic representation in dATL is specifically supported by language, as a unique mechanism of representing (abstract) semantic space, beyond the sensory-derived semantic representations distributed in the other cortical regions."By 'specifically', do the authors mean that dATL is a solely language-based representation (and sensory-derived conceptual representation plays no role, even in the sighted), do they mean the dATL specifically (and not other brain regions) is involved in language-derived conceptual representation. Does 'supported', mean a supportive/auxiliary role, or does this mean that language is the basis upon which representations are formed? Does 'unique' imply that the dATL alone is language-derived, that it is alone in being language derived and independent of sensory-derived representations, that is the unique representation of truly abstract knowledge?I would encourage the authors to, as much as possible, clearly lay out their claims, as this is of genuine interest (rather than just 'softening the language').

We appreciate this comment very much. We have made the following revisions to explicitly address these important issues.

We added a full paragraph in the Discussion to clarify the specificity regarding the anatomical regions (i.e., relative to other brain regions) and the information contents (i.e., relative to semantic structures derived from nonlinguistic experiences) (page 14): “Two types of specificity are to be clarified – anatomical specificity and information specificity. First, is this group effect specific to the dATL, relative to other brain regions? We do not have evidence for such a strong region-specificity. We did not observe a significant group-by-ROI interaction in either the RSA or the univariate analyses. That is, the group differences were not significantly stronger in the dATL than in the other semantic regions being analyzed (IFG and pMTG). We are thus not claiming that the dATL is the only region that derives semantic representation from language experience, but choose to focus the following discussion on this region because of the robust positive effects here. Second, is the dATL semantic representation specifically derived from language experience and not other (nonlinguistic) sensory experiences?.…” (see also response to comment 2 below).

We have also revised the Introduction to more explicitly explain the potential cognitive contribution of language to semantic knowledge and the rationale behind our study (page 2): “Language experiences may contribute to semantic development beyond sensory experiences by facilitating or modulating categorizations by the nature of labelling (i.e., words) to the sensory experiences, and/or by constructing semantic relations based on various types of word relations (Gelman and Roberts, 2017; Perszyk and Waxman, 2018; Unger and Fisher, 2021). Are such cognitive contributions manifested by modulating the neural representations of the sensory-derived semantic spaces, or by also formulating neural representations that specialize to represent knowledge derived from language experience (i.e., not nonlinguistic sensory experience)?”; (page 6): “We aim to test the neural system representing the language-derived semantic representations, beyond the sensory-derived semantic representations (Bi, 2021; Wang et al., 2020).”

We have also carefully updated the wording in other parts of the manuscript to make the relevant statements more precise.

Comment 2I do get the sense that the authors are proposing that dATL may be purely language-derived representation not grounded in or in any derived from sensory experience. It seems possible that this can be the case, as shown in the congenitally blind population, but this does not indicate that it is always the case. It is very possible that in the absence of sensory experience, dATL representations can continue to form, but it is simultaneously possible that when this sensory experience is present, more grounded processes also contribute. Are the authors arguing that the present result shows that this is not the case? Please clarify.

We appreciate this suggestion to clarify the discussion of this important issue. We have added explicit discussions as follows (page 14-15): “Is the dATL semantic representation specifically derived from language experience and not other (nonlinguistic) sensory experiences? The manipulation of the current study--the two groups of deaf signers--varies on the language exposure while matching on the sensory experiences (see below), and thus did not test the presence or absence of sensory-derived semantic representations. Inferences could be drawn in combination with the previous studies that focused on the manipulation of sensory experiences by studying visual knowledge in congenitally blind subjects. There, it was reported that the blind and sighted had comparable semantic information encoding in the RSA analyses (Wang et al., 2020); in terms of univariate effects (abstractness or color concept adaptation), deprivation of sensory experiences did not reduce, but actually tended to enhance the effects here (Bottini et al., 2020; Striem-Amit et al., 2018). That is, evidence from congenitally blind studies does not support the additional sensory-derived semantic encoding here. While drawing negative conclusions is always difficult, we reason that it is parsimonious, based on available data, to propose that dATL’s contribution to semantic encoding is specific to language, and not sensory-derived representations.”

Comment 3The main empirical emphasis of this work and the authors' general thesis is the specificity or particular 'peak' importance of the dATL. The group difference in the RSA and in the univariate analysis of the abstract>concrete effect in dATL do indeed support the importance of dATL. However, the failure of these effects to reach significance in the other abstract>concrete ROIs (pMTG, IFG) does not indicate that these effects are different between dATL and (e.g.) pMTG.Ideally, ROI by group interactions would be performed on the ROI data in Figure 2B and Figure 3A to show that dATL is statistically different from the other ROIs. I appreciate that this level of statistical vigour is often not met in our field but considering the importance and interesting nature of the authors' claims, I feel it would strongly improve the quality of the results here.If the ROI by effect interactions are not significant, then a clear caution about over-interpreting the absence of effects of pMTG and IFG could be included in the discussion.

We appreciate this comment very much. We added two-way ANOVA, with ROIs as the within-subject factor and group as the between-subject factor. The ROI-by-group interaction did not reach significance for either RSA (*F_(2,74)_* = 1.59, *p* = 0.21) or univariate abstractness (*F_(2,74)_* = 0.50, *p* = 0.61) effects (pages 9 and 12). We have revised the statistical description and relevant discussions accordingly, making sure to not claim for true negatives in pMTG/IFG or that the dATL is “anatomically specific”, being the only region that derives semantic representation from language experience.

Explicit clarifications included “We do not have evidence for such a strong region-specificity. We did not observe a significant group-by-ROI interaction in either the RSA or the univariate analyses. That is, the group difference was not significantly stronger in dATL than in the other semantic regions being analyzed (IFG and pMTG). We are thus not claiming that the dATL is the only region that derives semantic representation from language experience, but choose to focus the following discussion on this region because of the robust positive effects here.” (page 14)”

Comment 4Similarly, the searchlight analysis (lines 220-229) reports the presence of effects in native signers and the absence of effects in later signers (while there are no significant differences between the groups). This may also encourage the reader to over-interpret the absence of effects. The inclusion of the searchlight is clearly motivated, in that it shows that group differences are not apparent outside the ROIs analysed in the previous section. It would be good if the authors could emphasise more strongly the absence of effects outside of the ROIs and caution against the over-interpretation of the absence of effects in delayed signers given the absence of significant differences between delayed and native signers.

Thank you for this helpful advice. Following this suggestion, we added, after the report of the whole-brain map results: “Of course, such a thresholded map does not indicate true negatives in the delayed singer group”, which was followed by the emphasis on the group difference tests, summarized by the implications of the whole brain searchlight analyses “Together, group differences are not apparent outside the ROIs analyzed in the previous section.” (pages 9-10)

Comment 5Relating to comment 4, if we are to take the descriptive route, it seems there is comparable evidence implicating the left AG/TPJ in language-derived representations in native but not delayed signers (Figure 3C). Might a (post hoc) ROI analysis indicate comparable effects of delayed language acquisition in this region? How would the potential generalisability of this effect impact the authors' beliefs about dATL?

Following this comment, we carried out RSA in an additional set of language ROIs that included both ATL and AG/TPJ (Fedorenko et al., 2010) to further validate the effects in the ATL and to examine the potential group differences in the AG/TPJ (pages 10-11): “(2) Types of ROIs: To validate whether the dATL semantic reduction in delayed signers depends on particular ATL definitions and to explore potential group differences in other language-sensitive regions beyond the ROIs we localized, we performed the RSA in a commonly used language mask (contrasting intact sentences with nonword lists) (Fedorenko et al., 2010). As shown in Figure 4, again we observed significant group differences in the ATL (Welch’s *t*_33.1_ = 3.71, two-tailed *p* = 7.53 x 10^-4^, Hedges’ g = 1.18), which also survived the Bonferroni correction. Other language-sensitive regions did not achieve significance, with the tendency for the same direction of semantic encoding reduction (*p*s >.065, uncorrected). Two-way ANOVA showed a significant main effect of group (*F_(1, 37)_* = 6.80, *p* = .013) and no significant ROI-by-group interaction (*F_(5, 185)_* = 0.823, *p* = .535), indicating that delayed L1 acquisition resulted in widespread reduced semantic representations in the language regions, with the effects in the ATL consistently robust.” These results led us to explicitly acknowledge that we do not have evidence or claim for the strong region-specificity in the dATL.

Comment 6Delayed language acquisition could have potentially wide-reaching effects on cortical development. I appreciate that participants were matched on some criteria (e.g. self-report sign proficiency, lexical decision) but do the authors have reason to be sure that reported imaging differences are not a non-specific result of broad cortical changes? One supportive data point I can think of is the difference between dATL and IFG but perhaps there is information in the literature. It would be good if the authors could lay these out.

Thank you for the helpful suggestion. Indeed, previous studies have reported that delayed L1 acquisition associates with anatomical alterations. We have now reviewed such findings and considered our results in light of those observations in the Introduction: “Anatomical alterations in regions typically recruited in language tasks – reduced cortical volume in the left inferior frontal region, reduced cortical thickness in the left posterior middle temporal region, and reduced fractional anisotropy values in the left arcuate fasciculus – were also reported in signers with delayed L1 acquisition (Cheng et al., 2023, 2019).” (page 4)

And the Discussion: “The current results also have implications for the role of early language exposure in neurodevelopment more generally. Previous studies have reported delayed L1 acquisition associates with reduced cortical volume in the left inferior frontal region, reduced cortical thickness in the left posterior middle temporal region, and the reduced fractional anisotropy values in the left arcuate fasciculus that structurally connects the two regions, and not ATL or its related white matter tracts (Cheng et al., 2023, 2019). Our findings of semantic functional alterations in dATL are thus not easily attributable to broad anatomical changes associated with late L1 acquisition.” (page 17)

Comment 7Do the authors think that language-derived knowledge is related to all domains dATL? For instance, in congenitally blind individuals, do the authors think distinctions between a mountain range versus a city skyline would be present in dATL rather than PPA or RSA?

We appreciate the opportunity to clarify our positions. The general response to the hypothesized representation in dATL is shown in response to Comment 1. For the specific example here, a mountain range versus a city skyline may differ not only in language relational patterns, but also in nonvisual sensory aspects. Although congenitally blind individuals could not directly perceive such targets, they may still learn that a city skyline is comprised of more rectilinear or rectangular lines, and a mountain range may have more curved lines, or other types of compositional properties, from direct or indirect language descriptions. Such componential shape knowledge may be derived from sensory experiences (e.g., tactile), which could be represented in PPA/RSC. Thus it is difficult to make strong predictions about the absence of sensory-derived representations and we restricted our discussion to the main manipulation here (effects of language experience).